# Multi-ancestry meta-analysis of keloids uncovers novel susceptibility loci in diverse populations

Catherine A. Greene [1,2], Gabrielle Hampton[1], James Jaworski[1,3], Megan M. Shuey [1,4], Atlas Khan [5], Yuan Luo [6], Gail P. Jarvik [7], Bahram Namjou-Khales [8], Todd L. Edwards [3,12], Digna R. Velez Edwards [2,4,9,12] ✉ & Jacklyn N. Hellwege [1,9,10,11,12] ✉

Keloids are raised scars that grow beyond original wound boundaries, resulting in pain and disfigurement. Reasons for keloid development are not well-understood, and current treatment options are limited. Keloids are more likely to occur in darker-skinned individuals of African and Asian descent than in Europeans. We performed a genome-wide association study (GWAS) examining keloid risk across and within continental ancestry groups, incorporating 7837 cases and 1,593,009 controls. We detected 26 loci in the multi-ancestry analysis, 12 of which replicated in an independent dataset. Heritability estimates were 6%, 21%, and 34% for the European, East Asian, and African ancestry analyses, respectively. Genetically predicted gene expression and colocalization analyses identified 27 gene-tissue pairs, nine in skin and fibroblasts. Pathway analyses implicated integrin signaling and upstream regulators involved in cancer, fibrosis, and sex hormone signaling. This investigation nearly quintuples the number of keloid-associated risk loci, illuminating biological processes in keloid pathology.

Keloids are raised scars that expand beyond the original wound boundaries and encroach on the surrounding skin[1–3]. They are characterized by fibroproliferative derangement of the wound-healing process involving excessive deposition of collagen and overactive cell proliferation, though the precise etiology is not well-understood. The most common symptoms of keloids are pruritus (itchiness) and pain. More extensive or persistent keloids can cause disfigurement, and mobility issues may arise if keloids develop on joints. Notably, they have a tendency to recur after surgical excision and are frequently refractory to alternative treatments[4], necessitating an improved understanding of biological factors associated with susceptibility to excess scarring.

Keloids are most likely to affect darker-skinned individuals, particularly those of African or Asian descent. A sociodemographic study

[1]Vanderbilt Genetics Institute, Vanderbilt University Medical Center, Nashville, TN, USA. [2]Division of Quantitative and Clinical Sciences, Department of Obstetrics & Gynecology, Vanderbilt University Medical Center, Nashville, TN, USA. [3]Division of Epidemiology, Department of Medicine, Vanderbilt University Medical Center, Nashville, TN, USA. [4]Department of Biomedical Informatics, Vanderbilt University Medical Center, Nashville, TN, USA. [5]Division of Nephrology, Dept of Medicine, Vagelos College of Physicians & Surgeons, Columbia University, New York, NY, USA. [6]Department of Preventive Medicine (Biostatistics and Informatics), Northwestern University Feinberg School of Medicine, Chicago, IL, USA. [7]Departments of Medicine (Medical Genetics) and Genome Sciences, University of Washington Medical Center, Seattle, WA, USA. [8]Center for Autoimmune Genomics and Etiology, Cincinnati Children's Hospital Medical Center (CCHMC), Cincinnati, OH, USA. [9]Vanderbilt Epidemiology Center, Vanderbilt University Medical Center, Nashville, TN, USA. [10]Division of Genetic Medicine, Department of Medicine, Vanderbilt University Medical Center, Nashville, TN, USA. [11]VA Tennessee Valley Healthcare System (626), Nashville, TN, USA. [12]These authors jointly supervised this work: Todd L. Edwards, Digna R. Velez Edwards, Jacklyn N. Hellwege. ✉e-mail: digna.r.velez.edwards@vumc.org; jacklyn.hellwege@vumc.org

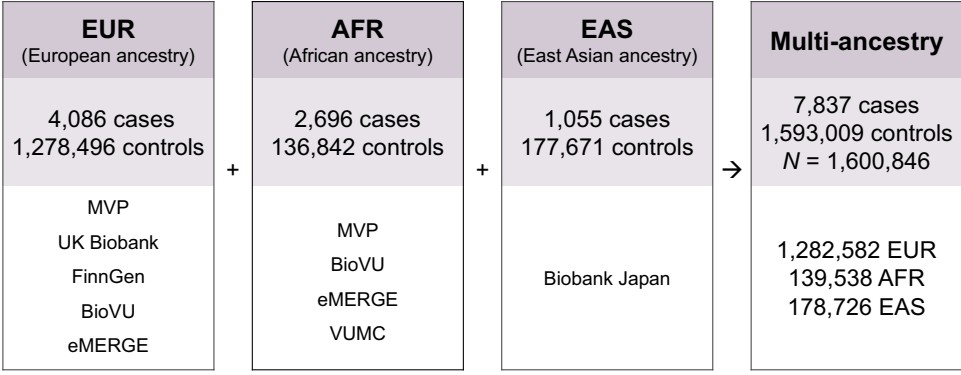

**Fig. 1 | Summary of populations for each meta-analysis.** EUR European ancestry meta-analysis, AFR African ancestry meta-analysis, EAS East Asian ancestry analysis, Multi-ancestry Multi-ancestry meta-analysis, combining all data sources. Cohort-specific information may be found in Table 1.

of keloids in the United States (US), with cases identified using structured and unstructured sources, documented a numerically higher proportion of Asian or Black patients with keloids[5]. In the US, keloids occur in about 1 in 30 Black individuals, approximately a 20-fold increase in risk compared to White individuals[6]. A recent study of excess scarring in the United Kingdom found prevalence estimates of 1.1%, 2.4%, and 0.4% for Asian, Black, and White patients, respectively[7]. Other fibroproliferative diseases such as hypertension, sarcoidosis, and uterine fibroids display similar increased prevalence, and it has been suggested these conditions share biology with keloids[8-10].

The limited genetic research conducted on keloids indicate they are a moderately heritable trait, as they can appear either sporadically or in people who have a family history. Family studies, however, have not identified a single genetic cause or mode of inheritance for keloids, fueling speculation that keloids are a complex trait with multiple susceptibility loci[11]. Seven genome-wide association studies (GWAS) since 2010, only two focused exclusively on keloids, have identified six distinct loci significantly associated with the risk of keloids[12-15]. However, these incorporated data from primarily European and East Asian ancestry populations; the earliest GWAS was a Japanese cohort study[12] while the others were derived from analyses in the UK Biobank[14,16,17], FinnGen[18], Biobank Japan[13,15], or meta-analyses of these data sources[19]. Only a recent analysis from the Million Veteran Program[20,21] included individuals of African ancestry, who have the greatest burden of disease. Our group previously conducted a whole-exome association and admixture mapping study in a Black population[22], and we now present a GWAS meta-analysis of keloids mapping risk in diverse populations.

We aimed to capitalize on the sample diversity and data availability of Electronic Health Record (EHR)-linked biobanks by conducting multi-ancestry analyses in BioVU and eMERGE, in coordination with other large-scale data sources including the UK Biobank, FinnGen, Biobank Japan, and the US Veterans Administration's Million Veteran Program (MVP). Because there are such striking health disparities in keloids susceptibility and severity, we conducted analyses stratified by ancestry group, depending on data availability, and present both multi-ancestry and ancestry-specific results. We also performed independent replication analyses in four ancestral populations from All of Us[23]. Finally, we performed enrichment and gene expression analyses to investigate the functional consequences of disease-associated genes and gain insight into the biology of keloid scars.

## Results

### GWAS Multi-ancestry Meta-analysis Results

We combined evidence of SNP-keloid associations through inverse variance-weighted fixed-effects meta-analyses, incorporating data from a total of 7837 cases and 1,593,009 controls (Fig. 1, Table 1) and limiting to common variants with minor allele frequency (MAF) ≥ 1%. We conducted meta-analyses both across and within ancestry groups,

### Table 1 | Cohort-specific sample sizes, case and population definitions

| Cohort | Case definition | Population definition | N Cases N Controls | | |
|---|---|---|---|---|---|
| | | | EUR | EAS | AFR |
| BioVU | Phecode 701.4 | EHR Race (NHW/NHB) | 163 47,047 | — | 111 10,605 |
| eMERGE | Phecode 701.4 | EHR Race (NHW/NHB) | 260 43,679 | — | 97 8395 |
| VUMC | ICD-9 code 701.4, clinical notes | EHR Race (NHB) | — | — | 122 356 |
| Million Veteran Program | Phecode 701.4 | Race/GIA* (HARE) | 2112 451,944 | — | 2366 117,486 |
| UK Biobank | Phecode 701.4 | GIA | 257 413,923 | — | — |
| FinnGen | ICD-10 code L91 | Finnish | 1294 321,903 | — | — |
| Biobank Japan | ICD-10 code L91 | Japanese | — | 1055 177,671 | — |

NHW=Non-Hispanic White; NHB=Non-Hispanic Black. *GIA=Genetically Inferred Ancestry.

enabling the identification of keloids genetic risk factors exhibiting either multi-ancestry or ancestry-specific effects. Lead SNPs detected in any of the meta-analyses were evaluated for independent replication in four ancestry groups from All of Us (3371 cases, 288,438 controls [Supplementary Table 1]).

The multi-ancestry analysis identified 26 significant ($p ≤ 5 × 10^{-8}$), conditionally independent autosomal loci with support from multiple datasets, plus one additional locus of interest on the X chromosome (Fig. 2, Table 2). Twenty of the 26 autosomal loci are previously undescribed genetic risk factors for keloids; we replicated five of six previously identified loci and found additional support for a locus recently featured in a publication from the Million Veteran Program (Supplementary Table 2)[21]. Of the ten keloid-associated SNPs listed in the GWAS Catalog (accessed December 2024)[24], seven were replicated at genome-wide significance in the multi-ancestry analysis. Two SNPs had suggestive ($p ≤ 1 × 10^{-5}$) evidence of association, and one SNP (rs1511412, *PRR23A*) lacked evidence of association outside the source Japanese population.

The most significant variant in the multi-ancestry analysis was rs10863683-C ($p = 1.52 × 10^{-79}$, Odds Ratio [OR] = 1.40 [95% Confidence Interval [CI] 1.35 − 1.45]), an intergenic variant near *LINC01705* (Table 2). This variant was originally detected in an East Asian population[13] but was also significant in each of our ancestry-specific analyses. It was the most significant variant in the European ancestry analysis ($p = 1.32 × 10^{-44}$,

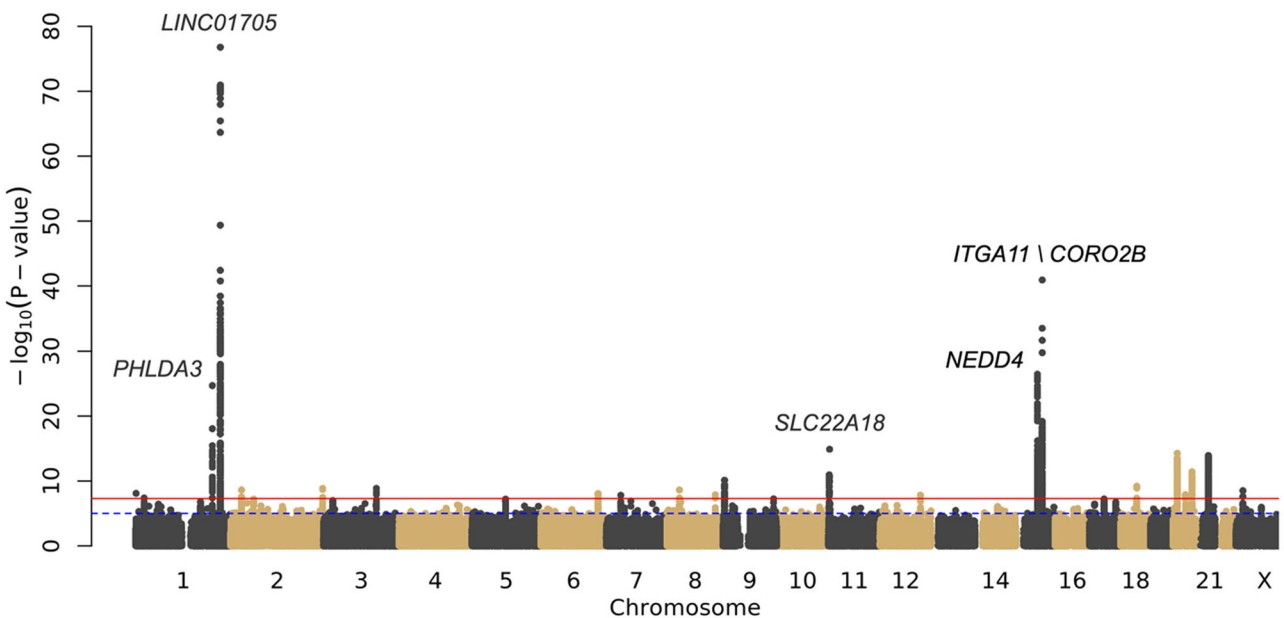

**Fig. 2 | Multi-ancestry meta-analysis of keloids replicates known genes and identifies previously unidentified keloids genomic risk loci.** Manhattan plot. The top five most significant loci are labeled with the nearest gene. The red line signifies the traditional GWAS significance threshold of $p < 5 \times 10^{-8}$ while the blue dotted line signifies the suggestive threshold ($p < 1 \times 10^{-5}$). Logistic regression statistical tests; multiple testing correction p-value threshold used ($5 \times 10^{-8}$). Plot created using the fastman R library.

**Table 2 | Independent lead SNPs in the multi-ancestry analysis, aligned to reflect increasing risk of developing keloids**

| SNP | Chr:BP | Nearest Gene | Effect | Ref | OR (95% CI) | Freq | P-value |
|---|---|---|---|---|---|---|---|
| rs12568930 | 1:22702231 | RP11-415K20.1 | C | T | 1.12 (1.07 — 1.16) | 0.24 | $5.92 \times 10^{-09}$ |
| rs35383942 | 1:201437832 | PHLDA3 | T | C | 1.50 (1.39 — 1.61) | 0.08 | $1.91 \times 10^{-25}$ |
| rs10863683 | 1:222251089 | LINC01705 | C | G | 1.41 (1.36 — 1.46) | 0.31 | $1.28 \times 10^{-79}$ |
| rs140707031 | 1:222291773 | LINC01705 | G | A | 1.58 (1.33 — 1.89) | 0.02 | $9.43 \times 10^{-09}$ |
| rs6726716 | 2:28381833 | BRE | A | G | 1.12 (1.08 — 1.17) | 0.78 | $8.18 \times 10^{-09}$ |
| rs12989123 | 2:241252201 | AC124861.2 | C | T | 1.16 (1.10 — 1.21) | 0.74 | $6.79 \times 10^{-10}$ |
| rs75826502 | 3:138837253 | MRPS22:BPESC1 | C | G | 1.99 (1.59 — 2.49) | 0.04 | $1.41 \times 10^{-09}$ |
| rs244755 | 5:88095785 | MEF2C | T | C | 1.10 (1.06 — 1.14) | 0.54 | $2.29 \times 10^{-08}$ |
| rs6906384 | 6:149664540 | TAB2 | G | A | 1.11 (1.07 — 1.15) | 0.56 | $5.36 \times 10^{-10}$ |
| rs2242026 | 7:37940286 | EPDR1 | T | C | 1.15 (1.09 — 1.20) | 0.13 | $7.74 \times 10^{-09}$ |
| rs2919386 | 8:32555685 | NRG1 | A | C | 1.22 (1.14 — 1.30) | 0.17 | $6.25 \times 10^{-10}$ |
| rs921721 | 8:126534536 | RP11-136O12.2 | T | A | 1.11 (1.07 — 1.15) | 0.57 | $1.93 \times 10^{-08}$ |
| rs6476838 | 9:4287190 | GLIS3 | T | C | 1.24 (1.16 — 1.32) | 0.80 | $1.20 \times 10^{-10}$ |
| rs686722 | 11:1891722 | LSP1 | C | T | 1.14 (1.10 — 1.18) | 0.63 | $1.00 \times 10^{-11}$ |
| rs76024540 | 11:2920108 | SLC22A18 | T | C | 1.44 (1.32 — 1.58) | 0.11 | $2.72 \times 10^{-18}$ |
| rs7297246 | 12:106119041 | CASC18 | A | G | 1.16 (1.10 — 1.22) | 0.82 | $4.48 \times 10^{-09}$ |
| rs11632096 | 15:56210499 | NEDD4 | G | A | 1.21 (1.17 — 1.26) | 0.32 | $1.11 \times 10^{-27}$ |
| rs34647667 | 15:68789866 | ITGA11 | T | G | 1.39 (1.32 — 1.45) | 0.76 | $1.86 \times 10^{-44}$ |
| rs77685836 | 18:42241036 | RP11-456K23.1 | C | G | 1.29 (1.19 — 1.40) | 0.93 | $2.07 \times 10^{-09}$ |
| rs2423510 | 20:10670079 | RP11-103J8.1 | T | A | 1.12 (1.08 — 1.16) | 0.54 | $5.74 \times 10^{-09}$ |
| rs140716753 | 20:11106003 | C20orf187 / LINC02871 | A | C | 1.68 (1.46 — 1.95) | 0.03 | $2.39 \times 10^{-14}$ |
| rs4239705 | 20:11242516 | RP4-734C18.1 | A | G | 1.15 (1.11 — 1.20) | 0.55 | $5.49 \times 10^{-15}$ |
| rs1205312 | 20:32849416 | ASIP:RP4-785G19.5 | A | G | 1.28 (1.17 — 1.39) | 0.08 | $4.25 \times 10^{-10}$ |
| rs6091310 | 20:49984404 | AL079339.1 | T | G | 1.13 (1.09 — 1.16) | 0.53 | $1.62 \times 10^{-11}$ |
| rs13051336 | 21:29816067 | AF131217.1 | A | G | 1.16 (1.11 — 1.20) | 0.48 | $1.60 \times 10^{-14}$ |
| rs2832056 | 21:30132507 | RNU6-872P | G | T | 1.12 (1.08 — 1.17) | 0.52 | $9.34 \times 10^{-10}$ |
| rs769545468* | X:20810164 | RP11-274G22.1 | A | G | 3.68 (2.39 — 5.65) | 0.01 | $3.06 \times 10^{-09}$ |

Nearest Gene reports the FUMA-mapped gene for the lead SNP, except for LINC01705 (mapped gene= RP11-400N13.1) to match previous keloid GWAS. SNP rsid, Chr Chromosome, BP Base Pair, Risk Risk Allele, Ref Reference Allele, OR (95% CI) Odds Ratio (95% Confidence Interval), effect conferred by the risk allele, Freq weighted average of risk allele frequencies across all datasets. *This SNP represented in only one dataset; the rest have multiple supporting datasets.

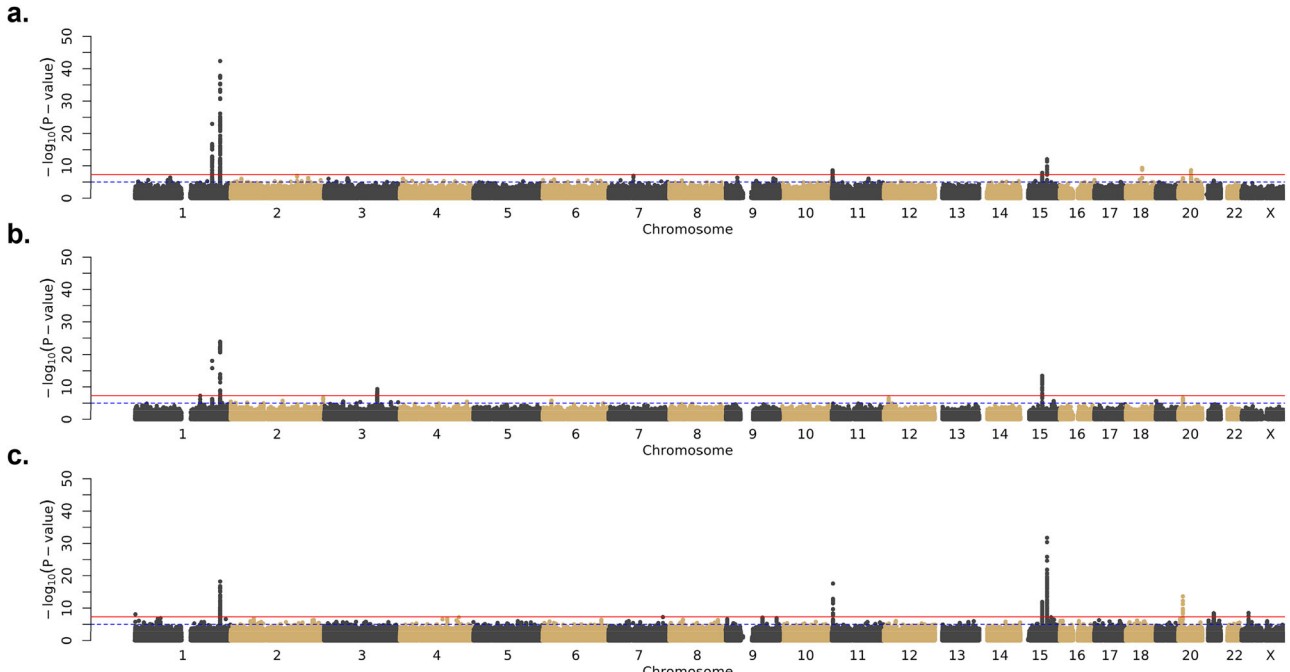

**Fig. 3 | Ancestry-specific keloid meta-analyses display consistency of major results across ancestry groups.** Manhattan plots. **a** European ancestry meta-analysis; **b** East Asian ancestry analysis; **c** African ancestry meta-analysis. Logistic regression statistical tests; multiple testing correction p-value threshold used ($5 \times 10^{-8}$). Plots created using the fastman R library.

OR = 1.39 [95% CI 1.32 − 1.45]) and the most significant variant at this locus in the African ancestry analysis ($p = 5.81 \times 10^{-19}$, OR = 1.34 [95% CI 1.26 − 1.43]) (Supplementary Data 1d, Supplementary Fig. 3). A nearby intronic variant, rs11293015-G, was the most significant result in the East Asian ancestry analysis ($p = 1.39 \times 10^{-24}$, OR = 0.60 [95% CI 0.54 − 0.66]) and may represent a conditionally independent locus in the Japanese population despite being in high Linkage Disequilibrium with rs10863683 in predominantly European populations (D' = 0.26, $r^2 = 0.046$ in Japanese; D' = 0.969, $r^2 = 0.823$ in European) (Supplementary Fig. 3)[25].

Another major locus in the multi-ancestry analysis was led by rs34647667-T ($p = 2.11 \times 10^{-44}$, OR = 1.39 [95% CI 1.33 − 1.46]), an intergenic variant located between *ITGA11* and *CORO2B* (Table 2). This SNP has the highest frequency in populations of African ancestry (gnomAD: MAF = 0.35 in African/African American compared to 0.15 in European [non-Finnish] and 0.04 in East Asian [Supplementary Table 3])[25] populations and was accordingly the most significant variant in the African ancestry-specific analysis ($p = 1.83 \times 10^{-32}$, OR = 1.48 [95% CI 1.39 − 1.58]). It also achieved significance in the European analysis ($p = 6.18 \times 10^{-14}$, OR = 1.30 [95% CI 1.21 − 1.38]), though the result at this locus was not significant in the East Asian analysis (rs34647667-T, $p = 0.015$) [Fig. 3, Supplementary Fig. 6].

**All of Us replication**

We sought to replicate our findings in an independent dataset, so we performed follow-up ancestry-stratified analyses with 3371 cases and 288,438 controls on the All of Us Researcher Workbench (Supplementary Table 1). Association analyses were conducted for available lead SNPs identified across the discovery meta-analyses, substituting for LD proxy SNPs where necessary (see Methods). In all, 22 SNPs displayed some evidence of replication ($p < 0.05$) in at least one of the four tested ancestry groups (European, East Asian, African, and Admixed American), with 12 meeting the significance threshold accounting for multiple testing ($p < 0.00128$, Supplementary Data 2).

The top result, rs10863683, was replicated in three populations (European [$p = 8.14 \times 10^{-11}$], African [$p = 2.42 \times 10^{-8}$], and Admixed American [$5.86 \times 10^{-7}$]) from All of Us, with some preliminary support from the East Asian analysis ($p = 0.011$). In the Admixed American / Latino analysis, seven SNPs had evidence of replication, but only two were significant (rs10863683, $p = 5.86 \times 10^{-7}$; rs11293015, $p = 6.42 \times 10^{-4}$). Both variants map to *LINC01705* but are in moderate LD (D' = 0.894, $r^2 = 0.5$) in Admixed American populations. Notably, rs11293015 was also replicated in the East Asian analysis ($p = 7.18 \times 10^{-3}$), providing some evidence for an independent locus in this population. The top previously unidentified result, rs34647667, was replicated in two populations (European [$p = 1.08 \times 10^{-7}$] and African [$p = 6.76 \times 10^{-14}$]). Another result, rs140716753 (*C20orf187 / LINC02871*), was only significant in the African ancestry discovery analysis but additionally had some evidence of replication in both the African ($p = 3.34 \times 10^{-13}$) and Admixed American ($p = 0.038$) replication analyses. Admixed American / Latino populations have not previously been included in keloids GWAS, and the replication confirms that at least some of the identified genetic risk factors for keloids also confer risk in this understudied population.

**LDSC Results**

We used Linkage Disequilibrium Score Regression (LDSC) to identify potential test statistic inflation and calculate the intercept for each analysis. Our multi-ancestry LDSC Intercept was 1.03, indicating that we do not have substantive confounding by population stratification (Supplementary Fig. 1, Supplementary Table 4). We also utilized LDSC to estimate SNP-based heritability of keloids from our ancestry-stratified GWAS summary statistics. Heritability was estimated to be approximately 0.06 in the European ancestry sample, 0.19 in East Asian ancestry, and 0.34 in African ancestry (Fig. 4, Supplementary Table 4). Heritability of keloids was highest in African ancestry, reflecting the observed pattern of disease prevalence in different populations.

**Ancestry Comparison Results**

Major results across ancestry-specific analyses were broadly consistent, though the relative strength of SNP-keloid associations tended to vary (Fig. 3). For instance, the chromosome 1 consensus result described above, corresponding to the locus at *LINC01705*, was highly

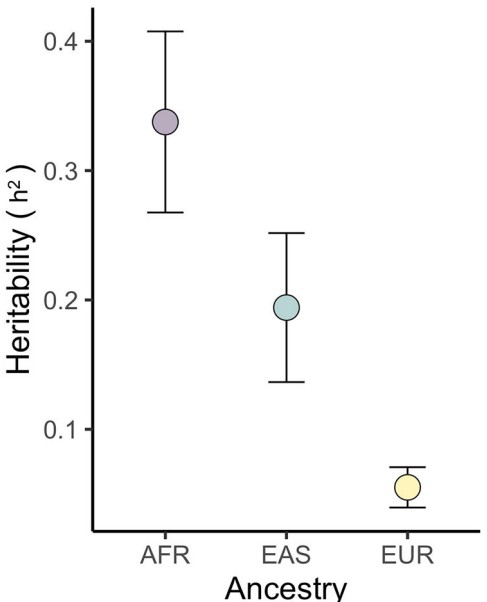

**Fig. 4 | SNP-based heritability (h²) estimates are highest in the African ancestry population and lowest in the European ancestry population.** Estimates were derived from LDSC analyses using ancestry-specific GWAS summary statistics. Data are presented as mean values +/- SE.

significant across all three ancestry-specific analyses but was superseded by other loci in the African and East Asian analyses (Fig. 3, Supplementary Fig. 3, Supplementary Table 3). A common result on chromosome 15 (rs11632096-A), mapping to *NEDD4*, was most significant in the East Asian ($p = 5.18 \times 10^{-13}$, OR = 0.71 [95% CI 0.65 − 0.78]) and African ($p = 3.14 \times 10^{-12}$, OR = 0.80 [95% CI 0.75 − 0.85]) analyses, but was less significant in the European analysis ($p = 7.07 \times 10^{-9}$, OR = 0.87 [95% CI 0.83 − 0.91]) (Fig. 3, Supplementary Fig. 6, Supplementary Table 3).

Many of the genetic risk loci for keloids also display heterogeneity among the ancestry-specific analyses. For example, the other strong result on chromosome 1, consisting of SNPs mapping to *PHLDA3*, is driven by the European (rs35383942-T, $p = 1.11 \times 10^{-23}$, OR = 1.49 [95% CI 1.38 − 1.61]) and East Asian analyses (rs192314256-C, $p = 8.74 \times 10^{-19}$, OR = 0.16 [95% CI 0.11 − 0.24]) (Supplementary Fig. 2). There is no peak at this locus in the African ancestry analysis (rs35383942-T, $p = 0.005$) though this discrepancy might be attributed to lower allele frequencies (MAF < 1%) in African/African American populations[25]. Further, the five distinct loci on chromosome 20 are driven by separate ancestry-specific analyses (Table 2). Three loci (rs2423510 [*JAG1*], rs6091310 [*NFATC2*], and rs4239705 [*RP4-734C18.1 / AL049649.1*]) were only significant in the multi-ancestry analysis. The variant rs2423510 was not significant or even suggestive ($p > 1 \times 10^{-5}$) in any of the ancestry-specific analyses; rs6091310 was only suggestive in the European analysis; and rs4239705 achieved suggestive significance in all three ancestry-specific analyses; but only the combined multi-ancestry evidence was sufficient to identify these keloid-associated loci. Two loci (rs140716753 [*C20orf187 / LINC02871*] and rs1205312 [*ASIP*]) were only significant in the African and European analyses, respectively, and may represent potential ancestry-specific genetic risk factors for keloids. Other ancestry-specific results include rs646315-T (*MRPS22*) on chromosome 3 in the East Asian analysis ($p = 4.63 \times 10^{-10}$, OR = 1.75 [95% CI 1.47 − 2.09]) (Supplementary Fig. 4); rs76024540-T (*SLC22A18 / SLC22A18AS*) on chromosome 11 in the African analysis ($p = 2.54 \times 10^{-18}$, OR = 1.47 [95% CI 1.35 − 1.61]); and rs769545468-A on chromosome X in the African analysis ($p = 4.05 \times 10^{-9}$, OR = 3.67 [95% CI 2.38 − 5.66]). Further sex-stratified analyses also identified this locus at suggestive significance in males ($p = 9.22 \times 10^{-8}$) but not females ($p = 0.020$).

## Fine-mapping of Genetic Signals

We used SuSiEx[26] to perform fine-mapping across ancestries, leveraging summary statistics from the three ancestry-specific discovery analyses. Across 38 tested regions (representing unique autosomal lead SNPs), we detected 47 credible sets. Of these, eight (excluding duplicate results from overlapping regions) had a posterior inclusion probability (PIP) greater than 0.9. (Supplementary Data 3).

Only two SNPs, rs11118950-T (*DUSP10*) and rs34647667-T (*ITGA11*), were represented across all three populations. The locus containing rs11118950 was defined for SuSiEx using the lead SNP rs191467669 from the African ancestry analysis. In European populations, this potential causal SNP is in moderate LD with rs10863683, the most significant result from the multi-ancestry meta-analysis (D′ = 0.723, r² = 0.48). However, low LD between rs11118950 and rs10863683 in African populations (D′ = 0.08, r² = 0.003) suggests an independent causal effect in this group. The other multi-ancestry causal SNP, rs34647667, is the same as the lead SNP identified for this locus in the European and African ancestry analyses.

The remaining SNPs were ancestry-specific results, consisting mostly of lead SNPs with sufficiently high MAF in only one population. These putatively causal SNPs included rs192314256 (*PHLDA3*) and rs150584862 (*LAD1*) in East Asians; rs35383942 (*PHLDA3*) in Europeans; and rs191467669 (*HHIPL2*), rs76024540 (*JAG1*), and rs140716753 (*JAG1*) in Africans.

## Consistency of Genetic Architecture across Ancestry Groups

To more broadly examine alleles influencing keloids risk across multiple ancestry-specific analyses, we compared both effect sizes and allele frequencies across European, East Asian, and African populations for the lead SNPs identified in the multi-ancestry meta-analysis. The European and East Asian analyses had 19 lead SNPs in common; the European and African analyses had 22 lead SNPs in common; and the East Asian and African analyses had 19 lead SNPs in common (Table 3, Supplementary Table 5). Pairwise comparisons revealed moderate to strong correlations for both effect sizes (Supplementary Fig. 7) and allele frequencies (Supplementary Fig. 8), though notably this analysis could not include five alleles represented by only one ancestry group (Table 3). We also performed the quantitative trait loci (QTL) sign test[27] using the 26 lead multi-ancestry SNPs to determine if allele frequency differences between populations constituted evidence of selection (Supplementary Data 1a). No significant ($p \leq 0.05$) differences were detected between the European and African analyses ($p = 0.14$), between the European and East Asian analyses ($p = 0.14$), or between the East Asian and African analyses ($p = 0.12$), suggesting that a greater number of keloid-associated alleles with frequency differences[28] would be needed to reject the null hypothesis of no selection.

## Functional Annotation of Keloid-Associated Genes

We utilized the online tool Functional Mapping and Annotation (FUMA) of GWAS[29] to map variants to genes; functionally annotate variants; and perform gene set, tissue, and pathway enrichment analyses. Utilizing the SNP2GENE module with our multi-ancestry results, we found that 119 autosomal genes were in proximity to regions containing significant keloid-associated variants. Functional annotation revealed enrichment for SNPs in non-coding regions, consistent with the findings of most previous GWAS[30]. Results were significantly enriched (E) for intronic noncoding (E = 2.09, $p = 1.89 \times 10^{-134}$) SNPs but were significantly depleted for intergenic (E = 0.742, $p = 6.92 \times 10^{-65}$) SNPs.

We also identified 12 nonsynonymous coding variants in LD (r² > 0.1) with FUMA lead SNPs (Supplementary Data 4). Six SNPs achieved either suggestive or genome-wide significance and were mapped to *PHLDA3*, *TAB2/SUMO4*, *LSP1*, and *NEDD4*. Other nonsynonymous SNPs, though not significant in this study, map to genes previously associated with fibroproliferative diseases (e.g. *SFRP4*[31]) and may be of interest for future research.

**Table 3 | Allele frequencies for population-specific SNPs**

| SNP | Effect | Ref | GWAS | | gnomAD | |
| --- | --- | --- | --- | --- | --- | --- |
| | | | Population | Freq | Population | Freq |
| rs75826502 | C | G | EAS | 0.0430 | East Asian | 0.0113 |
| rs140707031 | G | A | AFR | 0.0237 | African/African American | 0.0177 |
| rs76024540 | T | C | AFR | 0.1082 | African/African American | 0.1188 |
| rs140716753 | A | C | AFR | 0.0332 | African/African American | 0.0356 |
| rs769545468 | A | G | AFR | 0.0105 | African/African American | 0.0124 |

The highest population-level allele frequency for each SNP is shown, though each of these are the only populations in gnomAD where the frequency surpasses 1%. Risk=Risk Allele; Ref=Reference Allele; Freq=Allele Frequency.

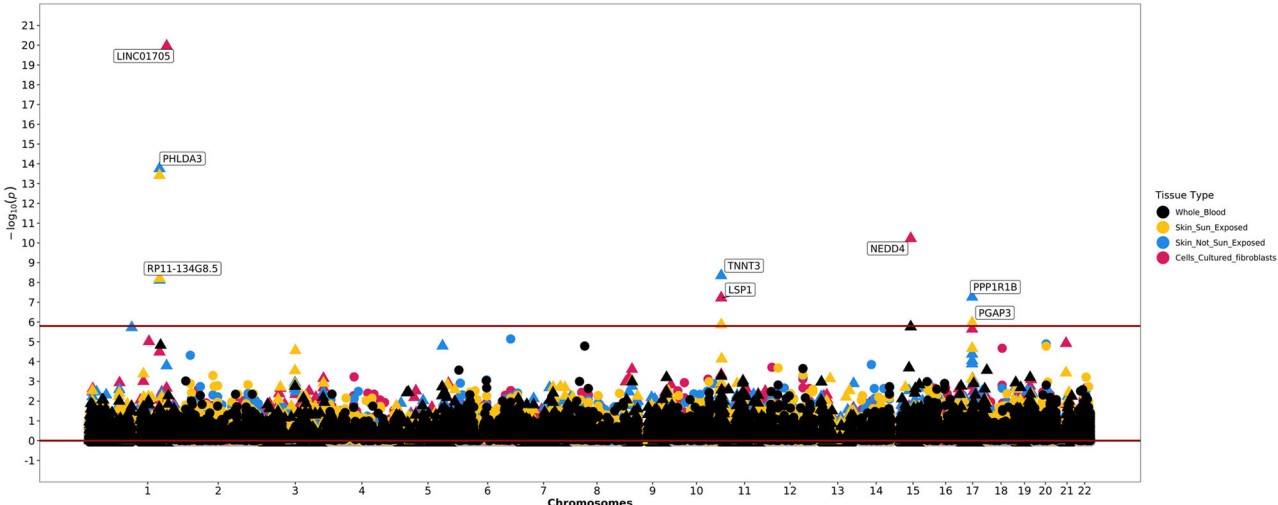

**Fig. 5 | Multi-ancestry GPGE analysis, restricted to keloid-relevant tissues.** Nine genes across four tissues are predicted to exhibit increased gene expression in response to keloids risk alleles. Two-sided Wald test; multiple testing correction significance threshold for the four keloid-relevant tissues = $p < 1.5 \times 10^{-6}$.

Predicted effects of significant SNPs were estimated through Combined Annotation Dependent Depletion (CADD)[32], RegulomeDB (RDB)[33], and Chromatine 15 interaction values[34] (E126, adult dermal fibroblasts) (Supplementary Data 5). In the multi-ancestry analysis, five of the 26 lead autosomal SNPs had CADD scores ≥ 12.37, considered the minimum value for pathogenic or highly deleterious SNPs[35]. Three SNPs (rs35383942 [*PHLDA3*], rs34647667 [*ITGA11/CORO2B*], and rs11632096 [*NEDD4*]) had CADD scores>20, placing them in the top 1% of pathogenic SNPs. The maximum CADD value (24.3) was for rs35383942 (*PHLDA3*) which also was likely to affect binding (RDB = 2a) and located within an active transcription start site (E126 = 1). Variants rs34647667 (CADD = 21.5, RDB = 3a, E126 = 14) and rs11632096 (CADD = 20.1, RDB = 2b, E126 = 7) were similarly predicted to affect binding and have impacts on transcription.

We conducted gene set enrichment analyses using FUMA's GEN-E2FUNC module, using the 119 mapped genes from the SNP2GENE module. The multi-ancestry tissue specificity analysis found that differentially expressed gene sets (GTEx v8 53 tissue types) characterizing the fallopian tubes ($p = 1.18 \times 10^{-4}$)[36], fibroblasts ($p = 3.61 \times 10^{-4}$), and arterial tissues were significantly up-regulated (Supplementary Data 6a). Significant (False Discovery Rate (FDR) < 0.05) gene sets included: genes co-amplified with *MYCN* in primary neuroblastoma tumors and genes within amplicon 20q11 with copy number variations in breast tumors in both Curated Gene Sets and Chemical and Genetic Perturbation Gene Sets; and genes involved in *ERBB4* signaling events in Curated Gene Sets and All Canonical Pathways. Ancestry-specific pathway enrichment results were sparse but included the amplicon 20q11 gene set in the European analysis and genomically imprinted genes in the African analysis. Several significant multi-ancestry results in Gene Ontology (GO) cellular components and GO molecular

functions related to muscle fibers and protein/lipid binding (Supplementary Data 6b). GWAS catalog reported traits had significant gene set enrichment for phenotypes including Dupuytren's disease, calcium level, prothrombin time, breast and oral cavity cancers, and body mass index – though this list consisted of more skin- and tissue-related traits such as keloid and lobe attachment in the ancestry-specific analyses (Supplementary Data 6c-e).

**Genetically Predicted Gene Expression (GPGE) Results**

We next investigated the potential functional effects that keloid-associated variants may have on gene expression in various tissues using S-PrediXcan, in 49 tissues from GTEx v8[37,38]. We detected 43 significant ($p < 1.7 \times 10^{-7}$) gene-tissue pairs in the multi-ancestry analysis, corresponding to 20 unique genes across 25 tissues (Supplementary Data 7). Results from three highly keloid-relevant tissues or cells (sun-exposed skin from the lower leg, not sun-exposed skin from the suprapubic region, and cultured fibroblasts) plus whole blood, included 11 significant ($p < 1.5 \times 10^{-6}$) gene-tissue pairs, with some associated genes in multiple keloid-relevant tissues (Fig. 5). The top result was for *LINC01705*; increased predicted expression of *LINC01705* in fibroblasts was associated with decreased risk of keloids ($p = 1.10 \times 10^{-20}$, OR = 0.65 [0.59 – 0.71]). Increased expression of both *PHLDA3* in not sun-exposed skin ($p = 1.74 \times 10^{-14}$, OR = 2.39 [1.91 – 2.99]) and *NEDD4* in fibroblasts ($p = 5.90 \times 10^{-11}$, OR = 1.91 [1.57 – 2.31]) were associated with increased risk of keloids. Increased predicted expression of *LSP1* in fibroblasts ($p = 6.01 \times 10^{-8}$, OR = 0.56 [0.46 – 0.69]) was associated with decreased risk of keloids. Although there was no significant GPGE result for *ITGA11*, the top result in the African ancestry analysis, we did observe a nearby association with *GLCE* in brain (amygdala, $p = 3.74 \times 10^{-19}$, OR = 0.36 [0.29 – 0.45]). We also identified

27 gene-tissue pairs (nine in keloid-relevant tissues) with significant GPGE results and high posterior probability (P$p$ > 0.9) of colocalization (Supplementary Data 8a).

The ancestry-specific GPGE analyses (all tissues) identified 41, eight, and nine significant gene-tissue pairs for the European, East Asian, and African analyses, respectively (Supplementary Data 7). Only three genes were significant across all three ancestry-specific analyses: *LINCO1705*, *NEDD4*, and *PRTG*. Other genes, notably *PHLDA3* and *RP11-134G8.5*, were significant in both the multi-ancestry and European analyses but not the East Asian and African analyses. Across the multi-ancestry and ancestry-specific analyses, we identified 22 unique genes with significant genetically predicted effects on gene expression. We used the union of the GWAS and GPGE gene lists as input for FUMA GENE2FUNC, which yielded significant (FDR < 0.05) enrichment of genes involved in myogenesis, cardiac development, and cancer pathways (Supplementary Data 6f).

### Ingenuity pathway analysis

We utilized QIAGEN's Ingenuity Pathway Analysis software[39] to determine potential upstream regulators and downstream biological functions of keloid-associated predicted gene expression, using nominally significant ($p$ < 0.05) GPGE results across the four meta-analyses. Twenty-five networks had significant enrichment of overlapping molecules in the multi-ancestry analysis, with top significant results including Carbohydrate Metabolism, Cell Cycle, Molecular Transport; Developmental Disorder, Gene Expression, Hereditary Disorder; and Cell Morphology, Cell-To-Cell Signaling and Interaction, RNA Post-Transcriptional Modification (Supplementary Data 9a). The top enriched canonical pathways included Generic Transcription Pathway (-log[p] = 7.73) and Axonal Guidance Signaling (-log[p] = 7.27), consistent across all analyses (Supplementary Data 9b, d). Other top pathways in the multi-ancestry analysis included Cardiac Hypertrophy Signaling (-log[p] = 5.64), Molecular Mechanisms of Cancer (-log[p] = 5.43), and Integrin Signaling (-log[p] = 4.88). The most significant upstream regulators, consisting of genes or other small molecules observed experimentally to affect expression of keloid-associated genes, included *HNF4A*, beta-estradiol, and dexamethasone across all analyses (Supplementary Data 9c, d). Other top upstream regulators included *TP53* in the multi-, East Asian, and African ancestry analyses; *ESR1* in the multi-, European, and African ancestry analyses; and *TGFB1* in the European and East Asian ancestry analyses.

## Discussion

We present a multi-ancestry meta-analysis of keloid scars incorporating large, diverse genetic datasets, notably including populations with increased incidence of fibroproliferative disease. Both multi-ancestry and ancestry-stratified analyses were conducted to facilitate broad discovery of keloid-associated genomic risk loci and examine their biological impact through follow-up functional annotation, GPGE, and pathway enrichment analyses. Through our discovery meta-analysis, we replicated five known keloids risk loci and identified a further 20 loci. Of twenty-six total loci, 12 replicated in at least one ancestry group in an independent sample. We additionally described 22 unique genes with genetically predicted effects on gene expression and evaluated genes' roles in various biological processes and pathways that might contribute to disease. Of these 22 unique genes associated with keloid susceptibility across our GPGE analyses, 17 were previously found to be differentially expressed when comparing keloids and healthy skin or comparing subpopulations of cells contributing to keloid pathogenesis (Supplementary Data 10).

We observed stark differences in estimated SNP-based heritability among the three ancestry-specific analyses, with keloids being most heritable in African ancestry populations ($h^2$ = 0.34) and least heritable ($h^2$ = 0.06) in European ancestry populations. These results reflect the observed pattern of keloids prevalence and suggest increased genetic

susceptibility to keloids in some populations. Interestingly, previous work examining genetic differences represented in HapMap3 genotype data found that SNPs with significant variation among geographic populations were in proximity to genes influencing hair, skin, and eye color characteristics – including skin-related disorders[40]. Our approach for the heritability analysis included a standard pre-processing step to restrict to approximately 1.2 million common HapMap3 SNPs, with our keloid heritability results potentially reinforcing prior findings. This disparity in keloids genetic risk also provides some support for hypotheses suggesting their increased prevalence in African and Asian populations may be due to positive selection of fibroproliferative alleles[8,9], though we were unable to gather further supporting evidence through our comparisons of keloid effect allele frequencies via the QTL sign test.

In the multi-ancestry meta-analysis, we identified 119 genes in proximity to keloid-associated loci. We replicated associations at *LINCO1705*, *PHLDA3*, *MRPS22/ BPESC1*, *SLC22A18/SLC22A18AS*, and *NEDD4* and identified numerous additional keloid-associated genes with diverse functions. Here we focus primarily on autosomal results, as the X chromosome lead variant (rs769545468) identified in BioVU was not replicated in the All of Us analysis.

Many of the keloid-associated genes have functions related to tumor suppression. The top result, *LINCO1705*, has been proposed as a regulatory factor underlying tumorigenesis. Some functional work examining SNPs in strong LD with rs873549 (a previous keloid-associated variant)[12] observed an association between rs1348270 (located in an enhancer) and downregulation of *LINCO1705*[41]. This result was concordant with our GPGE finding that increased expression of *LINCO1705* in fibroblasts was associated with decreased risk of keloids. This downregulation in normal fibroblasts was also found to increase expression of collagens via the *BMP2* pathway, potentially contributing to keloid formation. In addition to the mechanism described above, overexpression of *LINCO1705* was recently found to enhance cell migration and proliferation in breast cancer via regulation of *TPR*[42]. It was also observed to be differentially expressed in colon cancer and positively correlated with two immunotherapy indicators, microsatellite instability and tumor mutational burden[43]. These findings may be particularly relevant in considering population-specific genetic effects. The recently identified[21] keloid locus *SLC22A18/ SLC22A18AS* acts as a tumor suppressor in various cancers, including breast cancer and colorectal cancer[44,45]. Disease-associated variants in this gene have previously been found to vary markedly in frequency between continental populations[46,47], with some variants affecting the proliferation, migration, and invasion of colon cancer cells[47]. The lead SNP at this locus (rs76024540) was also found to be putatively causal in our African ancestry fine-mapping analysis, suggesting that population-specific genetic variants in *SLC22A18* might ultimately impact wound healing. Genes which have previously been associated with cancer phenotypes in the GWAS Catalog include: *LSP1*[48,49], *BRE / BABAM2*[50], *ASIP*[51,52], *TAB2*[48,49], *NRG1*[53,54], *TRIB1AL*[55], *GLIS3*[56,57], and *LINCO2871*[58]. Supporting these gene-level associations, an observational study examining risk of cancer development in patients with keloids found an overall increased cancer risk (OR = 1.49) in cases compared with controls, including increased risk for skin cancer (relative risk = 1.73)[59].

Other dermatologic and/or fibroproliferative conditions were associated with many genes, offering potential insights into keloid development. *LSP1* modulates adhesion and migration in primary macrophages, playing an important role in matrix remodeling and degradation[60]. *LSP1* was previously shown to characterize a major fibroblast population regulating inflammation in normal human skin[61]. Additionally, animal studies found that absence of *LSP1* promotes accelerated skin wound healing[62] and alleviates asthmatic inflammation[63] via reduced recruitment of inflammatory cell types. The region around *NEDD4* was previously found to contain an

admixture mapping peak associated with keloid formation in African Americans, with the most significant result at *MYO1E*[22]. The keloid-associated gene *ITGA11*, located approximately one megabase downstream of *NEDD4*, encodes a collagen receptor and is involved in the regulation of the profibrotic TGFβ-signaling pathway[64]. Integrins are the primary receptors for extracellular matrix components. They function to promote cell adhesion, migration, and turnover, including the construction and rearrangement of collagen matrices[65,66]. They also mediate the formation of the invadosome, a specialized structure that facilitates cellular invasion and matrix degradation. Importantly, the invadosome is comprised of podosomes, actomyosin-based organelles regulated by *LSP1*[60]. *ITGA11* specifically has been identified as a key player in tumor dynamics, as it is expressed by cancer-associated fibroblasts within the tumor microenvironment; is associated with aggressive tumor phenotypes; and is known to be upregulated in different kinds of cancerous or fibrotic lesions[66]. *ITGA11* was the top result overall in the African ancestry analysis and was previously associated with various kinds of organ fibrosis[64], uterine fibroids[15,67,68] and Dupuytren's disease[69–71], an abnormal accumulation of fibrotic tissue resulting in contracture of the hand. We did not observe a significant GPGE result at *ITGA11*, though the nearby associations at *GLCE* might indicate effects may be attenuated due to shortcomings of the current gene expression model.

Other relevant GWAS catalog phenotypes for these keloid-associated genes include several skin cancers; skin and hair pigmentation traits; male pattern baldness; earlobe morphology, and rosacea. Previously associated fibroproliferative conditions include asthma; glaucoma; hypertension and other blood pressure traits; uterine fibroids; and Dupuytren's disease. Other associated conditions consisted of musculoskeletal traits such as osteoporosis and decreased bone mineral density, which have been found in cross-sectional studies to co-occur with keloids[7,72]. The proposed mechanism for *LINC01705* may also provide some rationale for this co-occurrence, as downregulation of *LINC01705* was observed to promote expression of chrondrocyte- and osteocyte-associated genes via activation of *BMP2*, which is important in cartilage and bone formation[41]. Additionally, other keloid-associated genes, especially those with functions related to extracellular matrix like *ITGA11*, are necessary for bone formation, repair, and maintenance[73].

We observed several results that were consistent across pathway analyses, notably *TP53* and *TGFβ*. These genes have important roles in signal transduction; cell growth and differentiation; cell proliferation and migration; cell cycle signaling; and apoptotic pathways; and have both been robustly associated with fibrosis and cancers[74,75]. Dexamethasone, a glucocorticoid that downregulates *VEGF* expression, is used to suppress angiogenic activity as a first-line treatment for keloids[76,77]. *ESR1*, an estrogen receptor, is potentially reflective of the sex-specific effects of keloids, as cases of worsening keloids in pregnancy or after puberty in females have been documented[78–80]. The most significant networks largely involved cellular signaling and cancer pathways, supporting pathway enrichment results obtained through the FUMA GENE2FUNC analyses.

There are several considerations that may affect interpretation of results. The most important of these is cohort composition, as population definitions varied among source datasets (Table 1). Some biobanks utilized methods to obtain genetically inferred ancestry, while others relied on stratification of biobank populations by EHR-reported race (i.e., Black and White) and ethnicity (non-Hispanic). While ancestry and race are correlated[23,81], they capture different information, with ancestry being a better predictor of genetic factors. Future research may lead to the refinement of effect sizes and/or confidence intervals for ancestry-specific results. Case/control definitions also varied slightly between datasets, with the ICD-9 code 701.4 ("Keloid scar") being somewhat more specific than the ICD-10

code L91 ("Hypertrophic disorders of skin"). Most datasets identified cases using either definition, compiled into Phecode 701.4 ("Keloid scar") (Table 1). This approach may have resulted in the inclusion of hypertrophic scars, a condition of lesser severity that is difficult to clinically distinguish from keloids and which cannot be excluded with current code-based definitions. Though there are known differences in the pathophysiology of keloids and hypertrophic scars, it is unclear if their etiology is entirely separate or if they exist on a continuum of dysregulated wound healing[82]. Differing sample sizes among ancestry groups, previously discussed in context of specific analyses, impact our ability to draw conclusions regarding common versus ancestry-specific impacts of genetic risk factors for keloids. Current gene expression models are also characterized by a deficiency of samples from populations disproportionately impacted by fibroproliferative disease. Future work will ideally incorporate more diverse samples or be enabled by studies directly examining keloid-affected tissue. Despite these considerations, we have described numerous additional genetic risk factors affecting excess scarring across different populations.

Through this large multi-ancestry meta-analysis of keloids risk, we identified several genomic risk loci contributing to keloid development. Some had evidence of impacts on gene expression in keloid-relevant tissues like skin and fibroblasts. Many of the genes identified have previously documented roles in fibrosis and wound healing, with dysregulation affecting cellular functions contributing to diverse disease phenotypes. Our findings also offer preliminary support for genetic risk factors that vary based on ancestry, with stark differences in heritability indicating heterogenous genetic susceptibility to keloid scarring depending on the ancestral population. Our discovery dataset was limited to European, East Asian, and African groups, but we were able to extend results to additional populations in the replication analysis. Six SNPs had evidence of replication in the Admixed American group, encouraging future work in affected populations. An expanded understanding of the genetic architecture of keloids may assist in the identification of alternative treatments for scar management. This investigation may also serve to encourage future functional research examining mechanisms of keloids and highlight opportunities for studies of fibroproliferative disease.

## Methods
### Ethics
This research complies with all relevant ethical regulations and was designated exempt non-human subjects research by the Vanderbilt University Medical Center Institutional Review Board, as all study data was de-identified.

### Study populations
The total multi-ancestry meta-analysis amounted to approximately 1.6 million individuals across seven source biobanks. UK Biobank[16], FinnGen[18], Biobank Japan[15], and the Million Veteran Program (MVP)[20] all were previously utilized in studies of keloids. The MVP comprised a little over 85% of the samples in the African-ancestry meta-analysis. The remainder, which included both BioVU and VUMC resources and the eMERGE Network, were more recent additions for use in keloids genetic discovery.

All datasets utilized code-based definitions for the identification of keloid cases (Table 1). BioVU, eMERGE, MVP, and UK Biobank used phecode 701.4 to identify cases. VUMC used ICD-9 code 701.4 in addition to clinical notes to identify cases[22]. FinnGen and Biobank Japan used ICD-10 code L91 to identify cases. The prevalence of keloids in each dataset is reflective of their relative rarity in most populations: 0.3% of the European ancestry samples, 0.6% of the East Asian ancestry samples, and 2% of the African ancestry samples were keloid cases.

## BioVU and eMERGE GWAS Analysis

BioVU is the DNA biorepository at Vanderbilt University Medical Center, linked to de-identified electronic health records in the Synthetic Derivative[83]. DNA derived from peripheral blood samples was genotyped on a custom Illumina Multi-Ethnic Genotyping Array (MEGA-ex; Illumina Inc., San Diego, CA, USA) platform and were imputed through Trans-Omics for Precision Medicine (TOPMed)[84]. The Electronic Medical Records and Genomics (eMERGE) network is a national network supported by the National Human Genome Research Institute that connects data from EHR-linked DNA biorepositories across the country for large-scale collaborative research efforts promoting genomic medicine[85–87]. eMERGE samples were genotyped using various platforms and were imputed through Haplotype Reference Consortium (HRC)[88]. For both BioVU and eMERGE, analyses were stratified according to EHR race. Quality control procedures were performed separately for non-Hispanic White and non-Hispanic Black individuals. Variants were limited to those with INFO scores>0.5, minor allele count>5, and above 95% genotyping rate. The phecode 701.4 (keloids and hypertrophic scars) was used to identify cases in both BioVU and eMERGE (Table 1). Association analyses were performed with plink2 (PLINK v2.00a2LM)[89], adjusted by sex and the top 10 PCs.

The chromosome X analysis was performed using the same cases and controls as the autosomal analysis. Minor allele count (MAC > 5) and genotyping rate (0.05) filters were applied, and covariates included sex and the first 10 PCs. We utilized –xchr-model 2, which codes male X chromosomes 0/2. Variants were limited to those with INFO scores>0.7 and MAF > 0.01. Sex-stratified analyses were conducted to follow up on significant results in the primary analysis.

## Multi-ancestry and ancestry-specific meta-analyses

We conducted fixed-effects inverse-weighted meta-analysis of keloids GWAS datasets using METAL software v(2011-3-25)[90]. The multi-ancestry meta-analysis utilized all datasets (European, African, and East Asian ancestry) from each data source. The European ancestry meta-analysis utilized datasets with European or White population descriptors, including the UK Biobank and FinnGen, as well as results from the MVP, BioVU, and eMERGE. The African-ancestry meta-analysis utilized datasets with African or Black population descriptors, including results from the MVP, BioVU, and eMERGE. We performed GWAS meta-analyses of keloids including variants with MAF ≤ 1%, first conducted with all datasets for the multi-ancestry-ancestry analysis, then also stratified by broad ancestry group. We used the traditional genome-wide significance threshold of $5.0 \times 10^{-8}$ and set the suggestive threshold at $1.0 \times 10^{-5}$.

## Replication

We leveraged the All of Us[23] resource to replicate our findings in a diverse, independent dataset. Keloid cases were identified in a manner equivalent to the other datasets, using ICD-9 code 701.4 and ICD-10 code L91.0. Ancestry groups with $N$ keloid cases > 100 included African/African American, American Admixed/Latino, East Asian, and European. Ancestry-specific sample sizes may be found in Supplementary Table 2.

We performed association analyses using plink2 (PLINK v2.00a2LM)[89] for each of the lead variants (where available, including the chromosome X variant rs769545468) identified across the multi-ancestry and ancestry-specific meta-analyses, adjusted for sex and the first 10 PCs. PCs were calculated separately for each ancestry group, and association analyses were conducted in an ancestry-stratified manner. Variants with MAF > 1% in the given population were retained for the replication. Chromosome X analyses were performed as described above for BioVU and eMERGE. If lead SNPs were not available, we searched for proxy SNPs in LD ($r^2 > 0.8$) with the lead SNP in the given population[91]. Nine of the lead SNPs were found to be monoallelic in non-target populations. (Supplementary Data 2). The significance threshold was set at 0.05/39 = 0.00128.

## Ancestry comparison

Variants were restricted to those achieving genome-wide significance in at least one analysis and were additionally limited to lead SNPs to avoid bias introduced by LD. Variants were aligned as needed using the EasyQC[92] R package prior to conducting effect size and allele frequency comparisons between pairs of ancestry-specific analyses. Allele frequencies were acquired from gnomAD[25], using European (non-Finnish) values for EUR; East Asian values for EAS; and African/African American values for AFR. Pearson's correlation was calculated for each pairwise comparison and are reported in Supplementary Figs. 7 & 8.

## Fine-mapping

We utilized SuSiEx[26] with our ancestry-specific discovery results to identify credible sets of potentially causal SNPs (posterior inclusion probability [PIP] > 0.90). We used unique lead SNPs from the multi-ancestry and ancestry-specific analyses ($N = 38$) to define regions spanning each SN$p \pm 250$ kb. Other settings included: up to five causal variants per set; maximum iterations of 50; minimum purity of 0.5; tolerance of 0.0001; and a minor allele frequency ≥ 0.005 for each SNP.

## FUMA

Summary statistics for the multi-ancestry and ancestry-specific GWAS meta-analyses were uploaded to the Functional Mapping and Annotation (FUMA)[29] SNP2GENE module at https://fuma.ctglab.nl/snp2gene. Except for the East Asian analysis, as it had only one contributing dataset, we restricted each analysis to variants with multiple contributing datasets (HetDf>0). MAGMA was enabled, and genomic risk loci were identified using an $r^2 > 0.1$ LD threshold. FUMA-mapped genes were forwarded to the GENE2FUNC module for gene set and pathway enrichment analyses, enabling all background genes.

## LDSC

We used Linkage Disequilibrium Score Regression[93] software (v1.0.1) to examine inflation and estimate heritability. We utilized publicly available LD scores for European and East Asian populations, and used a custom population-matched set of LD scores for the AFR set derived from BioVU MEGA data[84]. The intercepts are reported in Results and in Supplementary Table 4. The $\lambda_{GC}$ values from LDSC were 1.07 for multi-, 1.05 for European, 1.00 for East Asian, and 1.07 for African ancestry analyses. We also used LDSC to estimate SNP-based heritability for each ancestry-specific meta-analysis, using corresponding GWAS summary statistics. We estimated on the liability scale, using previously published approximate population prevalences[94] to conduct our analyses.

## Genome-wide complex trait analysis (GCTA)

Joint and conditional analysis was performed with GCTA (v1.93.0)[95]. We utilized the -cojo method and set the significance threshold at $5.0 \times 10^{-8}$, performing analyses per chromosome and combining results to form the set of jointly independent significant signals. In the multi-ancestry analysis, SNPs represented by only one dataset (except for the East Asian analysis, which consists only of summary statistics from Biobank Japan) were excluded to attain a conservative estimate of independent loci with multiple lines of supporting evidence. Loci were considered conditionally independent loci if: $p < 5.0 \times 10^{-8}$ in both the meta-analysis (p) and the conditional (p_cond) analysis; $-\log_{10}(p) / -\log_{10}(p\_cond) < 1.5$, less than a 1.5-fold difference between meta- and conditional analyses; and the lead SNP was not in LD ($r^2 > 0.1$) with any other lead SNPs[96].

## Genetically predicted gene expression (GPGE)

We investigated the gene expression effects of keloid-associated variants with S-PrediXcan (v0.7.1)[37,97] using 49 tissues from GTEx v8[38]. The

threshold for statistical significance (all tissues) was defined as $1.8 \times 10^{-7}$, determined using the number of gene models and tissues analyzed. Results were further filtered to examine associations with predicted expression in particular tissues of interest, including sun-exposed skin from the lower leg, not sun-exposed skin from the suprapubic region, cell-cultured fibroblasts, and whole blood. The significance threshold for the four keloid-relevant tissues was $p < 1.5 \times 10^{-6}$. We additionally performed colocalization analyses to test the hypothesis that a single variant is responsible for both the GWAS signal and the predicted expression association identified in GPGE. Coloc (coloc R library v5.2.2.)[98], a Bayesian gene-level test, was used to compare the GWAS and GPGE summary statistics. A statistically significant GPGE result plus a posterior probability of 90% (P$p$ > 0.90) or greater was considered strong evidence of colocalization.

### IPA

GPGE summary statistics were filtered to results achieving nominal significance ($p < 0.05$), then analyzed using the core analysis function in Ingenuity Pathway Analysis (IPA) software (Qiagen)[39]. IPA was utilized for the multi-ancestry and ancestry-specific analyses, with results for networks, pathways, and upstream regulators ordered by enrichment p-value.

### Reporting summary

Further information on research design is available in the Nature Portfolio Reporting Summary linked to this article.

## Data availability

Summary statistics for the multi-ancestry and ancestry-specific meta-analyses (European and African) have been deposited in the GWAS Catalog under accession numbers GCST90652487, GCST90652488, and GCST90652489. Study-specific summary statistics for BioBank Japan [https://www.pheweb.jp/pheno/Keloid] and FinnGen [https://r8.finngen.fi/pheno/L12_HYPETROPHICSCAR] are available at their respective web portals. Summary statistics for MVP are available under dbGaP study accession phs002453.v1.p1 [https://ftp.ncbi.nlm.nih.gov/dbgap/studies/phs002453/analyses/HARE/]. UKBB data access can be requested [https://www.ukbiobank.ac.uk/enable-your-research/apply-for-access]. BioVU [https://victr.vumc.org/how-to-use-biovu/] and eMERGE [https://emerge-network.org/collaborate/] data also require approved access, which can be requested at their respective links. The predicted expression models [https://predictdb.org/post/2021/07/21/gtex-v8-models-on-eqtl-and-sqtl/] used are publicly available. The other data generated in this study are provided in the Supplementary Data files.

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

## Acknowledgements

CAG was supported by TL1TR002244. GH was supported by T32GM080178. MMS and JNH were supported by K12AR084232 (Principal Investigator: DRVE). DRVE was supported by R21AR067938. The eMERGE Network is supported by numerous grants from the NHGRI. including U01HG004438 (CIDR) and U01HG004424 (the Broad Institute). BioVU is supported by institutional funding, the 1S10RR025141-01 instrumentation award, and by the CTSA grant UL1TR000445. The authors acknowledge the UK Biobank, FinnGen, Biobank Japan, and the Million Veteran Program for graciously providing their GWAS summary statistics to the research community. The authors additionally thank MVP staff, researchers, volunteers, and especially participants who previously served their country in the military and have now generously agreed to enroll in the study. (See https://www.research.va.gov/mvp/ for more details). Thanks to all who contributed to this work, as well as to those who made their summary statistics publicly available.

## Author contributions

Conceptualization : C.A.G., T.L.E., D.R.V.E, J.N.H.; Data curation : C.A.G., T.L.E., D.R.V.E, J.N.H.; Formal analysis: C.A.G., J.J.; Funding acquisition: T.L.E., D.R.V.E.; Investigation: C.A.G., G.H., M.M.S.; Project administration: D.R.V.E., J.N.H.; Resources: A.K., Y.L., G.P.J., B.N.K.; Supervision: T.L.E., D.R.V.E., J.N.H.; Visualization: C.A.G.; Writing – original draft : C.A.G., T.L.E., D.R.V.E, J.N.H.; Writing – review and editing: C.A.G., G.H., J.J., M.M.S., A.K., Y.L., G.P.J., B.N.K., T.L.E., D.R.V.E., J.N.H.

## Competing interests

The authors declare no competing interests.
