## [Transparent Peer Review file · Nature Communications]

Multi-ancestry meta-analysis of keloids uncovers novel susceptibility loci in diverse populations

Corresponding Author: Dr Jacklyn Hellwege

Version 0:

Reviewer comments:

Reviewer #1

(Remarks to the Author)

This manuscript describes a multi-ancestry GWAS meta-analysis of keloid susceptibility in seven public datasets that variously include subjects of EUR, EAS, and AFR ancestry. In general, the work is thorough and well-performed, and the paper well-written. The results are generally interesting, though there are no “Eureka” discoveries. Overall, the study replicated five known keloid risk loci and identified 21 novel loci, with various levels of implication in a total 119 genes, many having functions related to tumor suppression. Perhaps the most interesting result is the estimation of SNP-based heritability from the ancestry-stratified GWAS summary statistics, which is very low in the EUR and EAS samples (0.06 and 0.19, respectively), versus 0.34 in the AFR sample, consistent with the considerably higher prevalence of keloids in subjects of AFR ancestry than in other groups. I wish the authors had presented a more in-depth analysis of the genetic drivers of these heritability differences, perhaps particularly including the five ancestry-exclusive alleles, and how these genetic drivers might relate to inferred functions. The authors note several appropriate caveats to their analyses and conclusions, most importantly difference in case definitions and ancestry definitions between the studies included in their meta-analysis; it might have been helpful to address the implications of these differences in more detail.

Specific points:

1. The authors should present a more in-depth analysis of the genetic drivers of the observed ancestry-related differences in SNP-based heritability. In particular, address the effects of the five ancestry-exclusive alleles included in the analysis. Also discuss whether these genetic drivers of apparent ancestry-related heritability differences might suggest possible functional differences between ancestry groups.
2. It would be appropriate for the authors to be more specific and detailed regarding both case-definition and ancestry definition/ascertainment differences among the studies included in their meta-analysis, and to present analyses of whether and how these differences might affect their results and conclusions.
3. The authors claim to have detected a “locus of interest on the X-chromosome”, tagged by rs769545468. Detection of association signals on the X-chromosome requires application of specialized methods; however, no such methods are presented in the Methods. Absent more extensive discussion of these analyses and their results, this reviewer is skeptical about this particular result.

Reviewer #2

(Remarks to the Author)

I recommend acceptance for publication. The authors should be commended for their original approach to investigating disparities in keloid susceptibility and severity using a GWAS meta-analysis. The identification of 27 significant SNPs which also replicated four of five previously published loci demonstrates the papers rigor. The multi-ancestry and ancestry specific loci results support clinical observed population differences in incidence. In addition, the results provide additional and more specific information on keloid susceptibility. Moreover, the identified 119 keloid associated gene variants provides important data for future functional and hypothesis generating research which was supported by the GPGE and subsequent IPA analyses identifying previously reported keloid associated genes and pathways. The keloid-associated gene variants and functional results could have been further strengthened with RNA-seq variant calling analysis of available keloid RNA-seq data (minor critique).

Reviewer #3

(Remarks to the Author)

Greene et al. report a multi-ancestry meta-analysis of genome-wide common variation associated with keloids leveraging data from multiple biobanks and population cohort. The authors report 21 new significantly associated loci and replicate other loci that have been previously reported. They also estimate the heritability of keloids in different ancestries and suggest genes involved in cancer and fibrosis to be involved in keloid development.

The study is well performed, the methodology is sound, and adds novel candidate genes that may help understand the pathophysiology of a poorly studied trait, such as keloids, especially given its higher prevalence in understudied non-European populations. A few comments and questions for the authors are below:

- In addition to African and East Asian ancestry populations, admixed Latin American individuals also have a higher prevalence of keloids. Therefore, given the authors' access to large biobank data linked to electronic health record information, why not include individuals from other ancestries where keloids also develop, such as Latin Americans? Even if in fewer numbers than other ancestries considered in the study, they could still be used and possibly informative for the larger multi-ancestry analysis. Similarly, given the previous work by these authors, why not include other East Asian populations that may have representation in the biobank studies used, such as Chinese individuals?

- While the functional annotation and pathway analyses are informative, the manuscript would be improved by expanding on the discussion of the actual functions and known biology of the novel proposed associated genes and how they could influence the pathophysiology of keloids.

Version 1:

Reviewer comments:

Reviewer #1

(Remarks to the Author)

Generally, the authors have been quite responsive to the concerns expressed. While they still don't seem to have carried out appropriate genetic analyses of the putative X-chromosomal association signal, they have de-emphasized this signal in light of its failure to replicate, largely allaying my concerns.

Reviewer #2

(Remarks to the Author)

The authors have addressed all concerns. I recommend acceptance.

Reviewer #3

(Remarks to the Author)

This reviewer's comments and questions have been addressed in the revision. I applaud the authors for their efforts in addressing the comments and including additional underrepresented populations in their study through a replication analysis using the All Of Us data. I believe this analysis has strengthened the authors' findings lending more credibility to the replicated loci. The manuscript is clearer and improved after review and the relevance of the analyses and findings are more evident.

We have addressed the reviewers' comments and edited the main text accordingly. Point-by-point responses to reviewer comments are below.

Reviewer #1 Comments:

This manuscript describes a multi-ancestry GWAS meta-analysis of keloid susceptibility in seven public datasets that variously include subjects of EUR, EAS, and AFR ancestry. In general, the work is thorough and well-performed, and the paper well-written. The results are generally interesting, though there are no “Eureka” discoveries. Overall, the study replicated five known keloid risk loci and identified 21 novel loci, with various levels of implication in a total 119 genes, many having functions related to tumor suppression. Perhaps the most interesting result is the estimation of SNP-based heritability from the ancestry-stratified GWAS summary statistics, which is very low in the EUR and EAS samples (0.06 and 0.19, respectively), versus 0.34 in the AFR sample, consistent with the considerably higher prevalence of keloids in subjects of AFR ancestry than in other groups. I wish the authors had presented a more in-depth analysis of the genetic drivers of these heritability differences, perhaps particularly including the five ancestry-exclusive alleles, and how these genetic drivers might relate to inferred functions. The authors note several appropriate caveats to their analyses and conclusions, most importantly difference in case definitions and ancestry definitions between the studies included in their meta-analysis; it might have been helpful to address the implications of these differences in more detail.

1. The authors should present a more in-depth analysis of the genetic drivers of the observed ancestry-related differences in SNP-based heritability. In particular, address the effects of the five ancestry-exclusive alleles included in the analysis. Also discuss whether these genetic drivers of apparent ancestry-related heritability differences might suggest possible functional differences between ancestry groups.

We thank the reviewer for their comments, particularly for their suggestion to include more information on ancestry-specific drivers of keloid susceptibility. To this end, we performed cross-ancestry fine-mapping with SuSiEx to further characterize patterns of associations across the three ancestral population groups (African, European, and East Asian). Regions for analysis were defined using lead SNPs from the multi-ancestry and ancestry-specific analyses, including the four autosomal SNPs with minor allele frequency > 0.01 in just one of the three source populations. We identified eight putatively causal SNPs, some of which appear to exert ancestry-specific effects due to frequency differences among populations. The detailed results from this analysis have been added to the main text (*Fine-mapping of Genetic Signals*, pg 9) and as a new Supplementary Table (ST8).

2. It would be appropriate for the authors to be more specific and detailed regarding both case-definition and ancestry definition/ascertainment differences among the studies included in their meta-analysis, and to present analyses of whether and how these differences might affect their results and conclusions.

Information regarding population and phenotype definitions used across studies can be found in Table 1, referenced in the *Considerations and Conclusions* section on pg 18. We have expanded our discussion of the potential implications of these differences.

3. The authors claim to have detected a “locus of interest on the X-chromosome”, tagged by rs769545468. Detection of association signals on the X-chromosome requires application of specialized methods; however, no such methods are presented in the Methods. Absent more extensive discussion of these analyses and their results, this reviewer is skeptical about this particular result.

We have updated the Methods section to include details pertaining to the chromosome X analyses (*BioVU and eMERGE GWAS Analysis*, pg 21). In response to a query from Reviewer #3 (see below), we performed independent replication of the 39 lead variants identified across the multi-ancestry and ancestry-specific meta-analyses. This included rs769545468 on chromosome X in 985 cases and 57,858 controls of African ancestry in the All of Us study. This variant did not replicate ($p=0.27$). We have also expanded slightly on the discussion of the X chromosome locus of interest (*Ancestry Comparison Results*, pg 8), though with caution to not over-interpret this non-replicated signal.

Reviewer #2 Comments:

I recommend acceptance for publication. The authors should be commended for their original approach to investigating disparities in keloid susceptibility and severity using a GWAS meta-analysis. The identification of 27 significant SNPs which also replicated four of five previously published loci demonstrates the papers rigor. The multi-ancestry and ancestry specific loci results support clinical observed population differences in incidence. In addition, the results provide additional and more specific information on keloid susceptibility. Moreover, the identified 119 keloid associated gene variants provides important data for future functional and hypothesis generating research which was supported by the GPGE and subsequent IPA analyses identifying previously reported keloid associated genes and pathways.

1. The keloid-associated gene variants and functional results could have been further strengthened with RNA-seq variant calling analysis of available keloid RNA-seq data (minor critique).

We thank the reviewer for their positive comments and appreciate their feedback. Although we were unable to directly access keloid RNA sequencing data for this study, we were able to conduct genetically predicted gene expression (GPGE) analyses that are reported in the manuscript (Genetically Predicted Gene Expression (GPGE) Results, pg 12-13). We have also further characterized our findings in the context of published RNA sequencing studies and expanded our discussion of genes (*Discussion*, pg 14-15). We have summarized this information into an additional Supplementary Table (ST15).

Reviewer #3 Comments:

Greene et al. report a multi-ancestry meta-analysis of genome-wide common variation associated with keloids leveraging data from multiple biobanks and population cohort. The authors report 21 new significantly associated loci and replicate other loci that have been previously reported. They also estimate the heritability of keloids in different ancestries and suggest genes involved in cancer and fibrosis to be involved in keloid development.

The study is well performed, the methodology is sound, and adds novel candidate genes that may help understand the pathophysiology of a poorly studied trait, such as keloids, especially given its higher prevalence in understudied non-European populations. A few comments and questions for the authors are below:

1. In addition to African and East Asian ancestry populations, admixed Latin American individuals also have a higher prevalence of keloids. Therefore, given the authors' access to large biobank data linked to electronic health record information, why not include individuals from other ancestries where keloids also develop, such as Latin Americans? Even if in fewer numbers than other ancestries considered in the study, they could still be used and possibly informative for the larger multi-ancestry analysis. Similarly, given the previous work by these authors, why not include other East Asian populations that may have representation in the biobank studies used, such as Chinese individuals?

Unfortunately, despite the overall large sample sizes in BioVU and eMERGE, only the European and African ancestry groups met our sample size requirement (roughly 100 keloids cases) for analysis, so additional analyses were not conducted for the other ancestry groups represented in these biobanks. However, in response to the reviewer's comments, we have since performed an independent replication study using the All of Us resource. All of Us intentionally recruits participants from diverse backgrounds, particularly those from historically understudied groups. This allowed us to examine associations between keloids and the identified lead SNPs in European (1,882 cases, 171,131 controls); East Asian (117 cases, 6,637); African (985 cases, 57,858 controls); and Admixed American/Latino (387 cases, 52,812) ancestry populations. 12 of the identified lead SNPs were replicated in at least one ancestry group, and seven SNPs had some evidence of replication ($p < 0.05$) in the Admixed American population. Methods (*Replication*, pg 22-23) and results (*All of Us Replication*, pg 7) for this replication have been added to the main text and the supplementary material (ST2 & ST6.).

2. While the functional annotation and pathway analyses are informative, the manuscript would be improved by expanding on the discussion of the actual functions and known biology of the novel proposed associated genes and how they could influence the pathophysiology of keloids.

We thank the reviewer for their suggestion. Further details on the functions and known biology of keloid-associated genes have been added to the discussion (*Discussion*, pg 15-18). We feel that this addition solidifies the involvement of several proposed processes in keloid pathogenesis, strengthening the manuscript overall.